# Leptin Mediated Pathways Stabilize Posttraumatic Insulin and Osteocalcin Patterns after Long Bone Fracture and Concomitant Traumatic Brain Injury and Thus Influence Fracture Healing in a Combined Murine Trauma Model

**DOI:** 10.3390/ijms21239144

**Published:** 2020-11-30

**Authors:** Anja Garbe, Frank Graef, Jessika Appelt, Katharina Schmidt-Bleek, Denise Jahn, Tim Lünnemann, Serafeim Tsitsilonis, Ricarda Seemann

**Affiliations:** 1Center for Musculoskeletal Surgery, Charité-Universitätsmedizin Berlin, Corporate Member of Freie Universität Berlin, Humboldt-Universität zu Berlin and Berlin Institute of Health, 13353 Berlin, Germany; frank.graef@charite.de (F.G.); jessika.appelt@charite.de (J.A.); denise.jahn@charite.de (D.J.); tim.luennemann@gmail.com (T.L.); serafeim.tsitsilonis@charite.de (S.T.); ricarda.seemann@charite.de (R.S.); 2Julius Wolff Institute for Biomechanics and Musculoskeletal Regeneration, Charité-Universitätsmedizin Berlin, Corporate Member of Freie Universität Berlin, Humboldt-Universität zu Berlin and Berlin Institute of Health, 13353 Berlin, Germany; katharina.schmidt-bleek@charite.de

**Keywords:** fracture healing, TBI, Insulin, osteocalcin, leptin, posttraumatic hormone household, bone hormone interaction

## Abstract

Recent studies on insulin, leptin, osteocalcin (OCN), and bone remodeling have evoked interest in the interdependence of bone formation and energy household. Accordingly, this study attempts to investigate trauma specific hormone changes in a murine trauma model and its influence on fracture healing. Thereunto 120 female wild type (WT) and leptin-deficient mice underwent either long bone fracture (Fx), traumatic brain injury (TBI), combined trauma (Combined), or neither of it and therefore served as controls (C). Blood samples were taken weekly after trauma and analyzed for insulin and OCN concentrations. Here, WT-mice with Fx and, moreover, with combined trauma showed a greater change in posttraumatic insulin and OCN levels than mice with TBI alone. In the case of leptin-deficiency, insulin changes were still increased after bony lesion, but the posttraumatic OCN was no longer trauma specific. Four weeks after trauma, hormone levels recovered to normal/basal line level in both mouse strains. Thus, WT- and leptin-deficient mice show a trauma specific hyperinsulinaemic stress reaction leading to a reduction in OCN synthesis and release. In WT-mice, this causes a disinhibition and acceleration of fracture healing after combined trauma. In leptin-deficiency, posttraumatic OCN changes are no longer specific and fracture healing is impaired regardless of the preceding trauma.

## 1. Introduction

The pathomechanism of the intriguing phenomenon of accelerated callus formation in polytrauma patients with concomitant traumatic brain injury (TBI), first described in 1964, remains unknown [1,2,3]. So far, cytokines, growth factors, regulatory proteins, mesenchymal stem cells, hormones, proteins, and even electrolytes have been investigated without a sufficient conclusion on the link between the central neuronal trauma and its bony effects [4]. Recently, the regulatory function of bone on energy homeostasis and the effects of metabolically active hormones on bone remodeling have been increasingly recognized.

Especially the discovery of the bone specific hormone osteocalcin (OCN) in 1975, together with more recent findings on the osteometabolic function of Insulin and Leptin, resulted in an increasing interest in the (neuro-) endocrine control of bone formation and the metabolic importance of the skeletal system as an endocrine organ [5,6,7,8,9,10,11].

Many experiments in this field already highlighted a close interplay of especially these three hormones, which together seem to play the central role in both–bone remodeling and energy homeostasis. Insulin, for example, showed to have an enormous influence on the skeletal system, as insulin-receptor-knock-out (KO)-mice suffered from impaired trabecular bone formation due to reduced osteoblast numbers, decreased bone formation rate and reduced osteoclast activation [12]. In addition, it was shown that intraosteoblast insulin signaling directly caused enhanced osteoclast activation, bone resorption and a concomitant increase in the release of this under- or un-γ-carboxylated OCN, Glu-OCN, from the extracellular matrix (ECM) [13,14]. As the hormone’s bioactive, soluble form, Glu-OCN executes its function in pancreatic β-cells, as well as in liver, bone and muscle cells. Here, it leads to the up-regulation of insulin production, release, and sensitivity and thus, closes a so-called insulin-OCN-fast-forward-loop, ultimately multiplying insulin’s metabolic as well as bone remodeling effects [11,15,16,17,18,19,20].

Similar experiments with leptin-KO-mice also showed this hormone’s positive peripheral effect on bone formation. However, unlike insulin, leptin also seems to have a negative central influence on bone remodeling via a hypothalamic relay and the activation of the sympathetic nervous system (SNS) [7,21,22]. In experiments with intracerebroventricular leptin injections, SNS-signaling leads to a β2A-receptor-stimulation in osteoblasts, which increases the expression of the embryonic-stem-cell-phosphatase (ESP) gene and causes a promotion of osteoclast differentiation and osteoblast testicular tyrosinphosphatase (OST-PTP) activity, which restrains the above described insulin-OCN-feed-forward-loop by a dephosphorylation of the osteoblast insulin receptor (for further details, see Figure 1 and Figure 2) [11,13,16,23,24,25].

In summary, various studies have shown a complex regulatory interplay of leptin, insulin, and OCN, which together balance bone formation with energy supply, storage, and expenditure aiming for an energy homeostasis in continuously shifting short and long term glucose levels. In this context, the skeletal system serves as a potent sensor, store, and effector in the interplay of energy offer and consumption [23].

In TBI and bone healing after polytrauma energy, homeostasis is severely disrupted and adequate hormone re-regulation urgently needed. Therefore, the aim of this study was to investigate the specific changes in posttraumatic insulin and OCN levels according to the preceding peripheral and/or central trauma and to further clarify the modulating role of leptin, by comparing the posttraumatic hormone changes of WT- and leptin-deficient mice.

## 2. Results

### 2.1. Insulin in WT-Mice

In our experiments on WT-mice, all animals in the three trauma groups, but especially mice after femur fracture and combined trauma, showed notably higher postoperative insulin values compared to the control group. Those values rose from two hours until two weeks after trauma, showing the overall highest values in animals after combined trauma (see Table 1). Three to four weeks after trauma, insulin levels slowly decreased to baseline and control values in all trauma groups, leaving values elevated the longest after Fx and combined trauma (see Table 1).

Accordingly, an analysis of variance (ANOVA) for further investigating trauma-specific differences showed highly significant group differences at two hours, two and three weeks after trauma (*p* = 0.000, 0.001 and 0.005). Post-hoc-analysis using Bonferroni correction confirmed significant and highly significant elevated insulin levels two hours after combined trauma compared to TBI and controls (*p* = 0.038 and <0.001). Two weeks after trauma, no group-specific significance could be traced, while all trauma groups had significantly and highly significantly elevated insulin values compared to control mice *p* = 0.047 (Fx), 0.004 (TBI), and 0.001 (combined trauma). Three weeks after trauma, the combined group again showed significantly the highest values even compared to the Fx and TBI group (*p* = 0.02 and 0.009) (see Figure 3).

### 2.2. Osteocalcin in WT-Mice

In contrast to insulin, in WT-mice, compared to controls, OCN levels were only slightly elevated two hours after Fx and even more distinct after combined trauma. One week later, OCN levels in these groups fell below control levels before they started to rise again from week two onwards. Surprisingly, TBI animals showed a small increase in OCN levels compared to controls two hours after surgery, which further extended until week three before starting to decrease again (See Table 2). Similar to the development in insulin levels, OCN values approximated normal and control values from week three after surgery, whereas normalization after TBI was delayed compared to the other trauma groups and combined trauma animals surpassed control and baseline levels from week three onwards (see Table 2).

For further investigation of potentially significant group differences, an ANOVA was conducted, which showed highly significant trauma specific values for one, two and three weeks after surgery (*p* = 0.006, *p* = 0.000, and *p* = 0.001). The Bonferroni correction confirmed significantly lower OCN values for combined trauma one week after surgery compared to TBI and controls (*p* = 0.012 and 0.023). Two weeks after trauma, only the highly significant difference between elevated OCN after TBI and decreased OCN after combined trauma remained detectable (*p* = 0.000), whereas all other groups showed no significant trauma specificity. Three weeks after surgery, OCN levels after TBI were furthermore significantly increased compared to OCN levels after combined trauma, but also after Fx (*p* = 0.022 and *p* = 0.000, respectively) before all groups again reached normal OCN values four weeks after trauma (see Figure 4).

### 2.3. Insulin in Leptin-Deficient Mice

In leptin-deficient mice, insulin levels were highly elevated in all groups and for all time points. Additionally, there was a postoperative rise in insulin values after trauma regardless of its specific nature. Especially leptin-deficient mice after Fx showed a rapid rise in insulin, reaching the highest values already one week after surgery. Comparable high values were found in the combined trauma and TBI group not before two weeks after surgery, when insulin values in Fx animals already started to decrease again. Fx and combined trauma groups fell below baseline and control levels four weeks after surgery, whereas TBI values only reached base line and control levels (see Table 3).

An ANOVA showed trauma-specific significances two, three, and four weeks after surgery (*p* = 0.008, 0.000, and 0.000). In Bonferroni correction, at two weeks, already decreasing insulin levels after Fx showed significance compared to the elevated insulin values after TBI (*p* = 0.034). Furthermore, with *p* = 0.061, at the same time there was a tendency towards significantly higher values after combined trauma compared to Fx only. Three weeks after surgery, TBI insulin values were significantly higher than those of all other groups (*p* = 0.000 (Fx), 0.000 (combined trauma), 0.005 (C)). Four weeks after trauma, insulin levels in the Fx group significantly fell below TBI and control levels (*p* = 0.039 and 0.000) and in the combined trauma group at least below control levels (*p* = 0.000) (see again Figure 2).

### 2.4. OCN in Leptin-Deficient Mice

In contrast to the insulin values, OCN levels in leptin-deficient mice were comparable to the ones in WT-mice. As already described for WT-mice OCN levels tended to drop after trauma, whereas in leptin-deficient mice also OCN levels after TBI showed a decrease instead of the elevated levels in WT-mice. In contrast to OCN in WT-mice, the lowest hormone levels for all trauma groups could be detected already one week after surgery before OCN values slowly recovered to base line but missing to reach control levels at week four. No trauma specific OCN development worth mentioning was to be observed (see Table 4).

In a subsequent ANOVA, group specific differences were detected 1, 2, and 4 weeks after surgery (*p* = 0.000 each), whereas Bonferroni correction could only detect significant OCN differences between trauma groups and controls (*p* = 0.000–0.04), but never between the different types of trauma (see again Figure 3).

### 2.5. WT-Versus Leptin-Deficient Mice

In comparison to WT-levels, general insulin values in leptin-deficient mice were elevated by 250. Furthermore, without leptin, the posttraumatic rise in insulin was about 8 times as high as in their WT counterparts. However, similar to WT-mice, in leptin-deficient specimens, insulin levels equally reached their maximum about 2–3 weeks after surgery. Because of the underlying extreme hyperinsulinaemia in leptin-deficiency, no direct statistic evaluation of similarities and differences between the mouse strains was possible as insulin values differ statistically significant at all times (see again Figure 2).

General OCN levels of both mouse strains were comparable. However, in the absence of leptin, there was a higher vulnerability of hormone homeostasis with a partly significant twofold pronounced posttraumatic OCN drop among leptin-deficient animals. Group-specific OCN reactions, like mentioned above for WT-mice, could not be detected in the case of leptin-deficiency. Additionally, the posttraumatic OCN drop was prolonged without leptin and recovery not seen before week 3 after surgery (see again Figure 3).

## 3. Discussion

As shown in this study, insulin and OCN levels directly react to injury, significantly differ according to the trauma inflicted, and seem to be mediated by leptin, as leptin-deficiency significantly changed posttraumatic hormone patterns in mice.

It was shown that central as well as extremity trauma lead to a reactive hyperinsulinaemia and a concomitant OCN drop in both mouse strains. In comparison, extremity and moreover combined trauma had a higher impact on posttraumatic hormone changes than TBI alone. Furthermore, hormonal vulnerability was shown to be significantly elevated in the absence of leptin, which underlines its otherwise regulatory function also described, for example, by Wei et al. 2008 and Wang et al. 2011 [26,27].

It was also shown that postoperative hyperinsulinaemia seems to be prolonged in persisting central energy requirement as during tissue regeneration after TBI and even more while competing with the skeletal system for adequate glucose supply during concomitant fracture healing [28,29]. This reaction seems to reach its maximum two to three weeks after trauma independent from mouse strain. From week three post-trauma onwards, stress and, accordingly, insulin concentrations finally recover to base line and control levels. This goes alongside with tissue recovery coming to an end, as was reproduced in weekly computer tomography (CT)-scans of the healing femora of all specimens in this study and reported before [30,31,32].

Finally, our study showed that the trauma specific OCN changes seen in WT-mice were no longer detectable in the absence of leptin, which again demonstrates this hormone’s essential role in posttraumatic bone and brain specific signaling.

Although it is known that the crosstalk of brain and bone is much more complex than described in the hormone signaling pathways above, applying those pathways to the present data may nevertheless help to describe how trauma specific hormone de- and rebalancing works and how concomitant TBI may induce an accelerated healing after long bone fracture. From this perspective and according to the signaling pathways described above, the posttraumatic insulin deregulation shown in our results may be the manifestation of a posttraumatic salvage brain-pull-mechanism, also described by Peters et al. 2004 in their “selfish brain theory”. Here, an immediate central nervous system (CNS) stress reaction leads to a central SNS-activation and induces a peripheral insulin resistance via the hypothalamus–pituitary–adrenal-corte*x*-axis to secure central glucose supply [28,29]. In reaction, ß-cells in the pancreas are stimulated to produce exceptionally more insulin, which could cause the posttraumatic hyperinsulinaemia seen in both mouse strains investigated.

As also described, a thus increased SNS-signaling leads to an increased OST-PTP-synthesis in osteoblasts, followed by higher OPG activity and a loss in osteoclast proliferation and activation. This, in turn, would lead to a reduced liberation of bioactive Glu-OCN from the ECM which again could be the reason for the posttraumatic OCN-drop also seen in this study [11,14].

Furthermore, the absence of a working Insulin-OCN-feed-forward-loop in an SNS-induced lack of free un-/under-carboxylated OCN, finally would help to end posttraumatic hyperinsulinaemia, marking the end of the energy homeostasis disruption.

How this hormone de- and rebalancing actually lead to an accelerated bone remodeling in the case of concomitant TBI remains still to be answered. However, as the results from this study show trauma-specific differences in the insulin and OCN household after injury and, moreover, the lack of a trauma-specific difference within the OCN-drop in the absence of leptin, these hormones seem to play an integral part in the brain–bone interaction. CT- and histological testing conducted in the mouse femora of all specimens included in this study showed increased bone remodeling in hyperinsulinaemic and hypoosteocalceamic WT-mice with combined trauma as well as an overall lack of bone remodeling in the absence of leptin in leptin-deficient mice [32,33].

Accordingly, Ducy et al. 1996 reported an increase in cortex thickness, bone mineral density (BMD), and bone mass (BM) in the absence of OCN, which they therefore denoted as a negative regulator of bone formation [15,25]. Recent animal studies on fracture healing showed high local and systemic leptin levels after bony trauma, exercising a direct, positive effect on bone formation via receptor mediated signaling in osteoblasts [27,34,35,36,37]. Turner et al. 2013 confirmed the importance of peripheral leptin for bone formation and additionally argued that the central leptin effect has been overestimated since Ducy et al. 2000 [7,38]. They even showed a positive bone remodeling effect after intracerebroventricular leptin infusion, which, in contrast to the results of Ducy et al. 2000, points to a possible positive central leptin effect [38].

To conclude, extreme and specific alterations in hormone concentration induced via bony and central trauma are reactions to homeostasis disruption and have an impact on bone remodeling. A stress induced rise in interleukin (IL)-1 and Il-6 levels as well as cortisol levels cause an elevated peripheral insulin resistance causing changes in bone remodeling. However, especially the post-traumatic rise in peripheral leptin conducts its positive and negative direct and indirect effects on bone remodeling peripherally and via its central relay. Considering that, additionally, after TBI the blood brain barrier (BBB) is deeply impaired and the otherwise limited transfer of leptin to the CNS via specific Ob-Ra-transporters is abolished, unphysiologically elevated systemic leptin concentrations may pass unimpeded to the CNS [39,40], where they are able to cause an elevated SNS-reaction and induce a TBI specific high OCN drop or support bone remodeling by still unknown positive central leptin effects so that bone healing is accelerated in the case of concomitant TBI until the BBB is restored as shown in Figure 5.

Our study highlights the deep rooted interaction of the central, endocrine, and musculoskeletal system and teaches us to recognize brain and bone in their dynamic and interventionalist nature. Its limitations include the relatively small sample size and the restriction to measurement of insulin and Osteocalcin levels due to the small volume of serum/plasma available. Other also interesting markers like leptin, blood glucose, inflammation markers, or bone remodeling markers therefore could not be analyzed in the study. Furthermore, as there is no ELISA for under-/uncarboxylated OCN, only total OCN could be measured and no data can be given on the actual hormone levels of bioactive OCN.

For the first time, it depicts injury dependent post-traumatic hormone changes in a customized mouse model, showing stress and healing reactions of insulin and OCN as well as the essential regulatory role of leptin. Therefore, the recent study uniquely confirms the complex interplay of hormones, CNS and bone in regulating energy distribution and healing and thus shows their role in times of homeostasis and its acute disruption after trauma. Set in context with other research in this field, it helps to further clarify the underlying mechanisms of bone formation, remodeling and healing.

Posttraumatic changes in hormone levels and the role of leptin probably need to be understood in the broader context of a physiological energy homeostasis, in which central and extremity trauma mark acute disruptions in an otherwise long-term balanced system, in which, with their high energy expenditure as well as their key role in maintaining a regular energy uptake, the brain and the skeletal system act as major players.

Future research is needed to investigate the exact insulin-, leptin-, and OCN-pathways in bone healing a well as to further depict bone’s regulatory role in energy homeostasis and the CNS’s role in bone remodeling. Here, innovative and integrating approaches like neuro-endocrinology and neuro-skeletal science are needed to answer the upcoming questions about the interdependency of brain and bone in energy homeostasis.

## 4. Materials and Methods

### 4.1. Experimental Design

All experiments were approved by local legal representative animal rights protection authorities (Landesamt für Gesundheit und Soziales, Berlin, G 0009/12, 4 June 2012) and carried out according to the policies and principles established by the Animal Welfare Act, the National Institutes of Health Guide for Care and Use of Laboratory Animals and the National Animal Welfare Guidelines.

All animals (120 female WT- mice and 120 female leptin-deficient mice, Charles River, Sulzfeld, Germany and Janvier, Saint Berthevin, France (for details see below)) were randomized into four groups: isolated fracture (Fx), isolated TBI (TBI), combined trauma (combined), and control (C) with femoral osteotomy and controlled cortical impact injury (CCII) serving as variables. Twenty animals per group and strain were sacrificed after 21 days and ten animals after 28 days.

### 4.2. Animal Care and Perioperative Management

Female C57/Black6N mice (Charles River, Sulzfeld, Germany, *n* = 120, age: 12–15 weeks, body weight: 22 ± 3 g) and B6.V-Lep-ob/JRj mice (Janvier, Saint Berthevin, France, *n* = 120, age: 11–15 weeks, body weight: 50 ± 5 g) were kept in groups of ten in standardized cages (type III “Euronorm”) with a twelve-hour light–darkness cycle and a controlled temperature of 20 ± 2 °C. Access to food and water was ad libitum. The animals were kept in the laboratory premises for at least one week prior to inclusion in the study, in order to allow for acclimatization and minimize stress.

For anesthesia, in all procedures, isoflurane 1.5 vol% (FORENE, Abbot, Wiesbaden, Germany) was used in 0.5 L/min N_2_O und 0.3 L/min O_2_. A heating pad (37 °C) prevented intraoperative hypothermia. Perioperative antibiotic prophylaxis was performed by a single shot subcutaneous injection of Clindamycin (0.02 mL). Skin was shaved and disinfected using 10% Braunol (B. Braun Melsungen AG, Melsungen, Germany). Subcutaneous application of Buprenorphine 0.1 mg/kg body weight (TEMGESIC, Reckitt Benckiser, Mannheim, Germany) ensured sufficient analgesia. Additionally, Tramadol 25 mg/L (TRAMAL, Gruenenthal, Aachen, Germany) was added to the drinking water (8 drops/250 mL of water) for three days postoperatively.

Animal sacrifice was performed after final heart puncture via atlanto-occipital dislocation under intraperitoneal anesthesia using 0.3 mL/kg body weight Medetomidin (DORMITOR, Orion Pharma, Bad Homburg, Germany) and 0.6 mL/kg body weight Ketamin 10% (KETAMIN, Actavis GmbH & Co. KG, Munich, Germany).

### 4.3. Surgical Procedures

For the Fx- and combined group, a standardized femoral osteotomy model stabilized with an external fixator was used [41,42]. A lateral longitudinal skin incision of 2 cm allowed for a mid-diaphyseal approach to the mouse femur. Dissection of the Fascia lata was followed by preparation of Musculus vastus lateralis and Musculus biceps femoris, carefully sparing the sciatic nerve. An external fixator (MouseExFix, RISystem, Davos, Switzerland) was mounted strictly parallel to the femur, positioning the pins perpendicularly to the longitudinal femoral axis. After rigid fixation, a 0.7-mm osteotomy was performed using a Gigli wire saw (RISystem, Davos, Switzerland). Wound closure was performed with Ethilon 6-0 suture (Ethicon, Johnson&Johnson, Norderstedt, Germany).

For the induction of TBI in the TBI and combined group, the standardized model of controlled cortical impact injury (CCII) was used [43]. Animals were mounted on a stereotactic device (Stoelting, Wood Dale, IL USA). A sagittal and temporal incision of the skin was followed by skin mobilization and preparation of the left temporal muscle. After craniotomy of the parieto-temporal region with a micro drill, a 7 × 7 mm bone window was lifted, carefully sparing the Dura mater. TBI was induced in a standardized manner (penetration depth 0.25 mm, impact velocity 3.5 m/s, contact duration 150 ms) with a pneumatic impactor device (AMS 201, AmScien Instruments, Richmond, VA, USA). The preserved piece of cranial bone was repositioned and fixed with dental cement (Hoffmann, Berlin, Germany) afterwards in order to simulate a closed brain injury. Wound closure was performed as described above.

### 4.4. Blood Sampling

Blood sampling took place via consecutive puncture of the Vena facialis (two drops of whole blood of each animal; 2 h, 2 and 3 weeks postoperatively) and via final heart puncture under intraperitoneal anesthesia (3 respectively 4 weeks postoperatively). Therefore, material for hormone analysis of each specimen was very limited.

Plasma was diluted with citrate in the ratio of 1:2 and centrifuged for 10 min at 1000 rpm. The cell plasma–citrate free supernatant was pipetted off and stored at −80 °C. Whole blood was centrifuged for 10 min at 1000 rpm. The cell free serum supernatant was stored at −80 °C.

### 4.5. Hormone Analysis

Quantitative insulin and OCN analysis were performed using mouse specific ELISA systems (Mouse Ultrasensitive Insulin ELISA, Version: 10 October 2012, Alpco Diagnostics, Salem, NH, USA, intra assay variation coefficient <9.3%, inter assay variation coefficient <11.5% and mouse osteocalcin ELISA, Immunotopics International, San Clemente, CA, USA, intra assay variation coefficient <3.7%, inter assay variation coefficient <6.1%). In spite of small sample volume (see above) pooling of samples was not necessary and data could be obtained for each specimen separately in single measurements.

### 4.6. Statistical Analysis

Statistical analysis was performed using SPSS 22 (IBM, Armonk, NY, USA). Continuous variables were expressed as means ± standard deviation (SD) or Median with its 25th and 75th percentile. For better homogeneity samples producing values more than 1.5 over the 75th or under the 25th percentile were defined as outliers and excluded from further statistical analysis. For testing of normal distribution the Kolmogorov–Smirnov or Saphiro–Wilks test was applied. Group-specific differences were tested using univariate analysis of variance (ANOVA) and Bonferroni correction. Differences were considered statistically significant if the null hypothesis could be rejected with >95% confidence (*p* < 0.05). Differences were considered highly significant if the null hypothesis could be rejected with >99% confidence (*p* < 0.01).

## Figures and Tables

**Figure 1 ijms-21-09144-f001:**
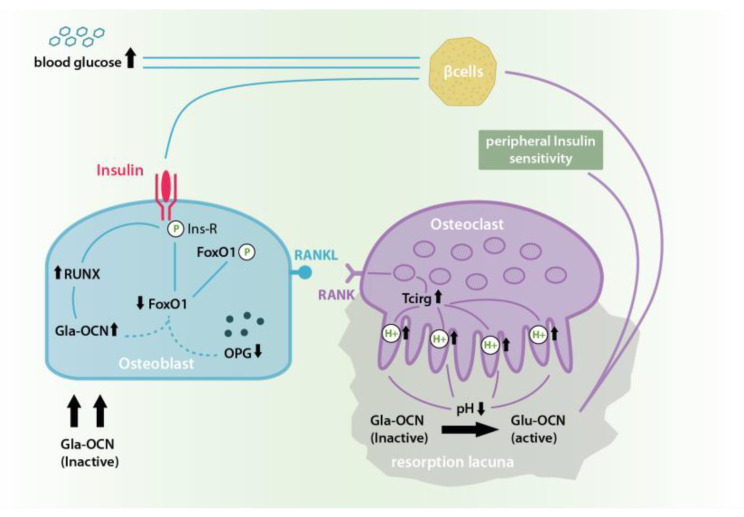
In the insulin-OCN-feed-forward-loop the binding of insulin to its receptor causes an Insulin receptor phosphorylation and activation which in turn causes the phosphorylation of Forkhead Box Protein O-1 (FoxO-1) and its inactivation. This reduces FoxO-1 levels, which otherwise would have inhibited Gla-OCN production and promoted osteosprotegrin (OPG) synthesis. At the same time, insulin signaling activates Runt-related transcription factor 2 (RUNX 2) which leads to a further increase in the expression and secretion of bio-inactive Gla-OCN. Insulin-dependent OPG reduction causes an increase in Receptor Activator of NF-κB/Receptor Activator of NF-κB-Ligand (RANK/RANKL) interaction and thus osteoclast differentiation and activation. This in turn leads to higher levels of bone resorption in acid resorption lacunae causing an increase in the pH dependent transformation of inactive Gla-OCN into under- or un-γ-carboxylated Glu-OCN and its release from the extracellular matrix (ECM). In its soluble and bioactive form Glu-OCN causes an improved peripheral insulin sensitivity as well as further β-cell activation and a consequent drop in blood glucose thus multiplying insulin’s physiological effects [13,14].

**Figure 2 ijms-21-09144-f002:**
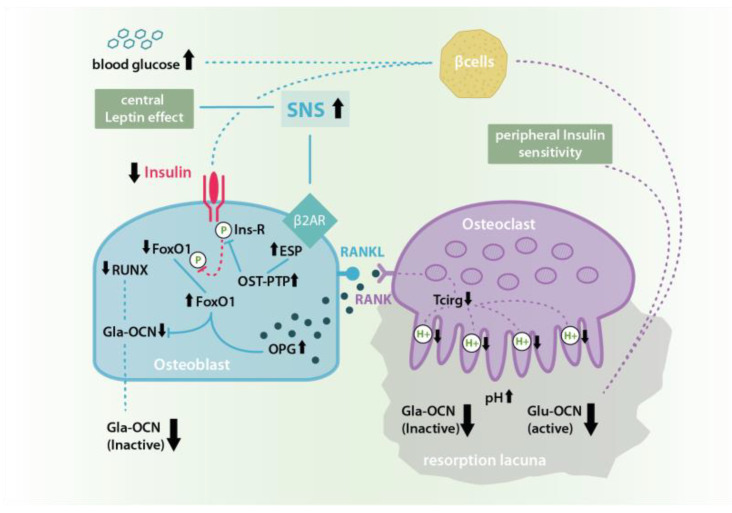
Leptin induced sympathetic nervous system (SNS) signaling at β2A-receptors in osteoblasts inhibits the Insulin-OCN-Feed-Forward-Loop by causing an increase in ESP gene expression and therefore osteoblast testicular tyrosinphosphatase (OST-PTP) synthesis. OST-PTP in turn dephosphorylates the insulin receptor as well as FoxO-1leading to missing Runt-related transcription factor 2 (RUNX) stimulation and Gla-OCN production as well as the disinhibition of OPG synthesis followed by a decrease of ostoeclast activation and pH-dependent Glu-OCN release from the ECM [11,13,16,23,24,25].

**Figure 3 ijms-21-09144-f003:**
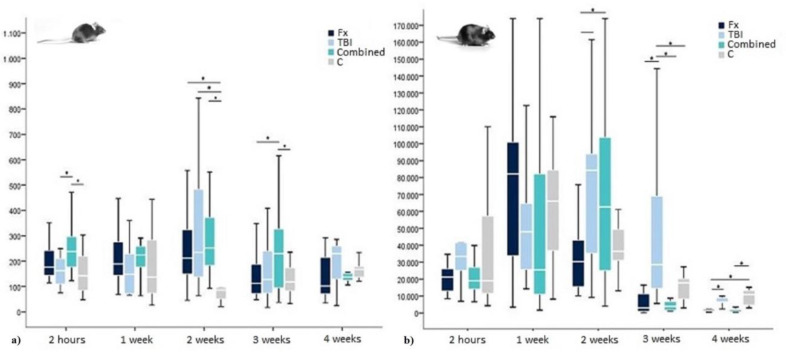
Insulin levels in (**a**) WT- and (**b**) leptin-deficient mice in pg/mL showing median values, 25th and 75th percentile as well as significant trauma specific differences 2 h, 1 week, 2, 3, and 4 weeks after surgery. * indicating a statistically significant difference between trauma groups (*p* < 0.05).

**Figure 4 ijms-21-09144-f004:**
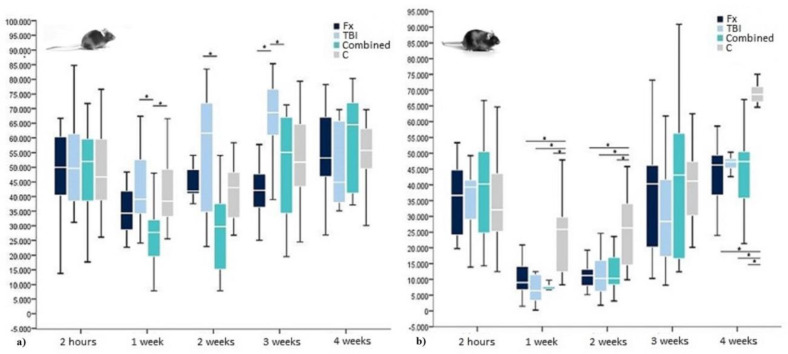
OCN levels within (**a**) WT- and (**b**) leptin-deficient mice in pg/mL showing median values, 25th and 75th percentile as well as significant trauma specific differences 2 h, 1 week, 2, 3, and 4 weeks after surgery. * indicating a statistically significant difference between trauma groups (*p* < 0.05).

**Figure 5 ijms-21-09144-f005:**
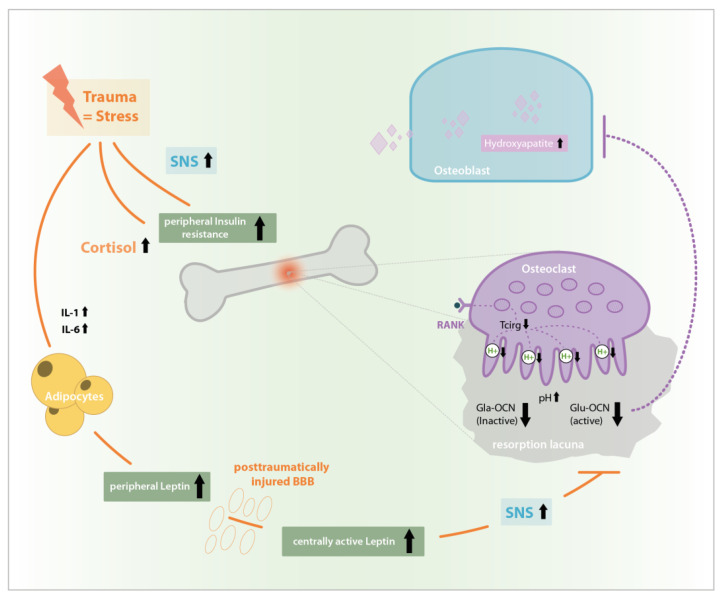
Stress induced rise in interleukin (IL)-1, IL-6, and cortisol levels as well as SNS activation causes peripheral insulin resistance, a Glu-OCN-drop and an increased leptin synthesis. In impaired blood brain barrier (BBB) after traumatic brain injury (TBI) central leptin levels and central effects are extremely increased, causing unphysiologic SNS-activation, Glu-OCN drop and disinhibition of bone formation after TBI which possibly causes the phenomenon of accelerated bone healing after combined bony and central trauma.

**Table 1 ijms-21-09144-t001:** Insulin levels in WT-mice in pg/mL showing median values and interquartile ranges from 2 h until 4 weeks after surgery.

		Fracture	TBI	Combined	Control
Insulin 2 Hours	Median in pg/mL	177	162	237	142
	Interquartile Range in pg/mL	106	115	137	146
Insulin 1 Week	Median in pg/mL	189	148.5	225	138
	Interquartile Range in pg/mL	138	159	84	240
Insulin 2 Weeks	Median in pg/mL	211.5	234	252	85.5
	Interquartile Range in pg/mL	176	423	210	71
Insulin 3 Weeks	Median in pg/mL	113	127.5	229.5	117.5
	Interquartile Range in pg/mL	124	171	241	98
Insulin 4 Weeks	Median in pg/mL	102	229	138	167
	Interquartile Range in pg/mL	180	153	33	47

**Table 2 ijms-21-09144-t002:** OCN levels in WT-mice in pg/mL showing median values and interquartile ranges from 2 h until 4 weeks after surgery.

		Fracture	TBI	Combined	Control
OCN 2 Hours	Median in pg/mL	50,064	49,638	51,891	46,701
	Interquartile Range in pg/mL	22,098	27,636	21,870	21,951
OCN 1 Week	Median in pg/mL	34,263	39,078	27,711	38,490
	Interquartile Range in pg/mL	14,703	24,588	14,965	16,536
OCN 2 Weeks	Median in pg/mL	41,892	61,596	29,733	43,104
	Interquartile Rangein pg/mL	10,686	44,613	23,728	16,818
OCN 3 Weeks	Median in pg/mL	42,093	68,655	55,104	51,717
	Interquartile Range in pg/mL	13,593	16,605	35,964	22,371
OCN 4 Weeks	Median in pg/mL	53,106	44,862	64,470	55,719
	Interquartile Rangein pg/mL	24,728	30,109	36,600	16,362

**Table 3 ijms-21-09144-t003:** Insulin levels in leptin-deficient mice in pg/mL showing median values and interquartile ranges from 2 h until 4 weeks after surgery.

		Fracture	TBI	Combined	Control
Insulin 2 Hours	Median in pg/mL	21,320	33,472	19,064	19,026
	Interquartile Range in pg/mL	14,484	26,935	16,096	49,030
Insulin 1 Week	Median in pg/mL	82,188	47,934	25,536	66,022
	Interquartile Range in pg/mL	84,972	44,307	73,852	53,700
Insulin 2 Weeks	Median in pg/mL	30,419	84,300	62,772	36,444
	Interquartile Range in pg/mL	31,299	62,324	88,050	24,174
Insulin 3 Weeks	Median in pg/mL	3058	28,572	3992	17,760
	Interquartile Range in pg/mL	11,615.5	74,414	6104	13,398
Insulin 4 Weeks	Median in pg/mL	1204	6396	2177.5	10,825
	Interquartile Range in pg/mL	1309.5	3407	1683.5	10,025

**Table 4 ijms-21-09144-t004:** OCN levels in leptin-deficient mice in pg/mL showing median values and interquartile ranges from 2 h until 4 weeks after surgery.

		Fracture	TBI	Combined	Control
OCN 2 Hours	Median in pg/mL	36,669	39,258	40,236	32,046
	Interquartile Range in pg/mL	21,219	19,158	28,836	22,068
OCN 1 Week	Median in pg/mL	9030	6416	7761	25,803
	Interquartile Range in pg/mL	9051	8339	1284	17,713.5
OCN 2 Weeks	Median in pg/mL	11,255	10,334	10,376	26,370
	Interquartile Range in pg/mL	5901	10,270	8772	20,022
OCN 3 Weeks	Median in pg/mL	40,365	28,419	43,098	41,238
	Interquartile Range in pg/mL	30,585	24,532.5	42,006	18,667.5
OCN 4 Weeks	Median in pg/mL	46,320	47,280	47,334	68,631
	Interquartile Range in pg/mL	14,170	4149	24,642	5623.5

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
