# Peer review of "Leptin Mediated Pathways Stabilize Posttraumatic Insulin and Osteocalcin Patterns after Long Bone Fracture and Concomitant Traumatic Brain Injury and Thus Influence Fracture Healing in a Combined Murine Trauma Model"

_ijms, 2020, doi:10.3390/ijms21239144_

Round 1
Reviewer 1 Report
Nice and original paper.
Have you look to bone cells while measuring these parameters? How the bone remodeling was in your mouse models? Are osteoclast active the same after different types of injury?Have you measured undercarboxylated or total osteoclacin?
Have you measured other bone markers (as CTX, P1NP, TRAcP).
Are inflammatory status monitored in same way?
Did you check other parameters of stress? (in 2019 Karsenty group described osteocalcin as involved in acute stress) Do you think your results could be influenced by this effect?
Why you analysed only female mice?
You have some histological picture of fracture healing in all the 5 condition? it will be nice to show differences also by pictures.
Minor revision: some mispelling in the text and I suggest to improve picture quality and probably also change some colors because everything is too blue and not of rapid comprehension.
Author Response
Dear editors and reviewers,
Thank you very much for your reply and the critical comments, remarks and suggestions on our manuscript entitled “Leptin mediated pathways stabilize posttraumatic Insulin and Osteocalcin patterns after long bone fracture and concomitant traumatic brain injury and thus influence fracture healing in a combined murine trauma model”.
After carefully considering all remarks and, wherever possible, revising the manuscript and figures as advised by the reviewers, we would like to re-submit the manuscript for publication in the International Journal of Molecular Sciences. The revised manuscript has been read, seen and approved by all authors.
Point-to-Point-Reply to Reviewer 1:
- Have you look to bone cells while measuring these parameters? How the bone remodeling was in your mouse models? Are osteoclast active the same after different types of injury?
Thank you for this remark. Indeed we did perform histological staining and analysis and found that there is a statistical difference in callus formation and bone healing within the trauma groups and between the two mouse strains. We also did further histological stainings, e.g. for osteoclasts (TRAP-staining). You may find our data on this in the following article, which we also provided as reference to the recent paper:
Seemann, R.; Graef, F.; Garbe, A.; Keller, J.; Huang, F.; Duda, G.; Schmidt-Bleek, K.; Schaser, K.D.; Tsitsilonis, S. Leptin-deficiency eradicates the positive effect of traumatic brain injury on bone healing: histological analyses in a combined trauma mouse model. Journal of musculoskeletal & neuronal interactions 2018, 18, 32-41.
- Have you measured undercarboxylated or total osteoclacin?
Thank you very much, we absolutely agree on this to be important information. Due to the fact that there is no standard ELISA for undercarboxylated OCN on the market we only measured total OCN. We tried to mark this limit in the text as follows and added the following sentence to the manuscript (lines 302-305): “Furthermore, as there is no ELISA for under-/uncarboxylated OCN only total OCN could be measured and no data can be given on the actual hormone levels of bioactive OCN. Therefore, we only can make assumptions to the actual level of undercarboxylated OCN.”
- Have you measured other bone markers (as CTX, P1NP, TRAcP)? Are inflammatory status monitored in same way?
Due to the fact that there was only a very limited amount of blood for each specimenand time point (only 2 drops of whole blood were to be taken at each time point as repeated blood loss was restricted by animal welfare regulations), we were forced to limit our analysis to Insulin and OCN. Inflammation markers were not monitored or compared in our study, but this would be of great interest for future studies! We tried to highlight this limitation in the text and added the following sentence (line 300-302): “Other also interesting markers like Leptin, blood glucose, inflammation markers or bone remodeling markers therefore could not be analyzed in the study.”
- Did you check other parameters of stress? (in 2019 Karsenty group described osteocalcin as involved in acute stress) Do you think your results could be influenced by this effect?
Again, thank you very much for this comment and arousing this important point. There was a daily check up on animal health and behaviour. Animals suffering from pain or surgery associated complications were recognized by changes in food intake, movement, social behaviour, and/or physical appearance. Here no group/trauma specific differences in stress/pain/health issues could be detected.
- Why you analysed only female mice?
Thank you very much for this question. In the context of our study bone healing was analyzed using weekly micro-CT-scans in addition to the hormone analysis. Bone healing in young male mice is known to be accellerated compared to female mice, so we were afraid to miss crucial trauma specific differences if we were only to do one scan/blood sample a week. On the other hand we wanted to spare our animals the stress of more than one anaesthesia/scan/blood sample per week.Furthermore, male mice would have been supposed to be held in separate cages to prevent aggressive intragroup behaviour. However, this seperation, in fact, is known to cause additional animal stres, whic we wanted to prevent in our study and terefore decided to analyse only female mice.
- You have some histological picture of fracture healing in all the 5 condition? It will be nice to show differences also by pictures.
As histological pictures of fracture healing were already published in the following reference articles, we decided not to add them to the actual article in order not to overburden the reader with information on the already very complex issues.
Locher, R.J.; Lunnemann, T.; Garbe, A.; Schaser, K.; Schmidt-Bleek, K.; Duda, G.; Tsitsilonis, S. Traumatic brain injury and bone healing: radiographic and biomechanical analyses of bone formation and stability in a combined murine trauma model. Journal of musculoskeletal & neuronal interactions 2015, 15, 309-315
Graef, F.; Seemann, R.; Garbe, A.; Schmidt-Bleek, K.; Schaser, K.D.; Keller, J.; Duda, G.; Tsitsilonis, S. Impaired fracture healing with high non-union rates remains irreversible after traumatic brain injury in leptin-deficient mice. Journal of musculoskeletal & neuronal interactions 2017, 17, 78-85).
- Minor revision: some mispelling in the text and I suggest to improve picture quality and probably also change some colors because everything is too blue and not of rapid comprehension.
Thank you very much. The whole manuscript was checked or misspellings and we had a native speaker check the text. The pictures were revised and added in higher solution and quality.
Reviewer 2 Report
Dear colleagues,
many thanks for the interesting publication.
I only would suggest minor revisions as follows:
(1) Please provide images ina better quality.
(2) It might be interesting to conduct also a statistical analysis to measure intraindividual differences. This might show any variations of the different levels over time. Please include/comment.
(3) Please describe also the limitations of the present study.
Best wishes
Author Response
Dear editors and reviewers,
Thank you very much for your reply and the critical comments, remarks and suggestions on our manuscript entitled “Leptin mediated pathways stabilize posttraumatic Insulin and Osteocalcin patterns after long bone fracture and concomitant traumatic brain injury and thus influence fracture healing in a combined murine trauma model”.
After carefully considering all remarks and, wherever possible, revising the manuscript and figures as advised by the reviewers, we would like to re-submit the manuscript for publication in the International Journal of Molecular Sciences. The revised manuscript has been read, seen and approved by all authors.
Point-to-Point-Reply to Reviewer 2:
- Please provide images in a better quality.
All images were revised and added in higher solution and quality.
- It might be interesting to conduct also a statistical analysis to measure intra-individual differences. This might show any variations of the different levels over time. Please include/comment.
Thank you for this important comment. We agree with the reviewer that an intra-individual analysis will provide further insights into hormone reactions to trauma as well as individual re- and destabilisation of hormone homeostasis. From animal and clinical studies we already know that there is a posttraumatic stress reaction with higher levels of blood glucose and insulin which is known in trauma surgery and acute care as permissive hyperglycaemia which occurs especially after multiple or major injury. Furthermore there exist studies on animals and in clinical setting describing changes in OCN levels after bony trauma and OCN is used in studies to monitor the quality fracture healing.
Therefore, given the fact that there is already literature available on this question, we decided to not include this analysis into our paper as our main focus was not to monitor hormone development over time, but to identify trauma specific differences in hormone levels via inter-individual analysis.
- Please describe also the limitations of the present study.
Thank you, we very much agree that it is important to rise the readers’ awareness to limitations of the present study. We carefully revised the paragraph in order to make it more clear to the reader and added the following sentences (lines 298-305): “Its limitations include the relatively small sample size and the restriction to measurement of Insulin and Osteocalcin levels due to the small volume of serum/plasma available. Other also interesting markers like Leptin, blood glucose, inflammation markers or bone remodeling markers therefore could not be analyzed in the study. Furthermore, as there is no ELISA for under-/uncarboxylated OCN only total OCN could be measured and no data can be given on the actual hormone levels of bioactive OCN. Therefore, we only can make assumptions to the actual level of undercarboxylated OCN.”
Round 2
Reviewer 1 Report
Thank you for your reply.
It is not sufficient the improve of picture. Osteoclast has a bad resolution. Change!
Probably it will be useful also to change the color of osteoblast to make a contrast with the black of word.
Author Response
Dear editors and reviewers,
Thank you again for your reply and the critical comments on our manuscript entitled “Leptin mediated pathways stabilize posttraumatic Insulin and Osteocalcin patterns after long bone fracture and concomitant traumatic brain injury and thus influence fracture healing in a combined murine trauma model”.
We changed our figures according to the good suggestions made by our reviewers and now would like to re-re-submit the manuscript for publication in the International Journal of Molecular Sciences. The revised manuscript has been read, seen and approved by all authors.
Point-to-Point-Reply to Reviewer 1:
- It is not sufficient the improve of picture. Osteoclast has a bad resolution. Change!
Probably it will be useful also to change the colour of osteoblast to make a contrast with the black of word.
Thank you for giving us feedback on the revised quality of our figures again. And thank you for your recommendation to change colours to make our visuals better understandable. We tried to improve quality and colour s so we hope our figures are now readable and much more comprehensive.